# The Influence of Movement on the Cerebrospinal Fluid Pressure of the American Alligator (*Alligator mississippiensis*)

**DOI:** 10.3390/biology11121702

**Published:** 2022-11-25

**Authors:** Bruce A. Young, Michael Cramberg

**Affiliations:** Department of Anatomy, Kirksville College of Osteopathic Medicine, Kirksville, MO 63501, USA

**Keywords:** neuroscience, fluid mechanics, reptile, dynamics, dura

## Abstract

**Simple Summary:**

The cerebrospinal fluid (CSF) supports, nourishes, and cleans the brain and spinal cord. Fulfilling these functions requires a certain dynamic of the CSF, it is constantly being produced (primarily via arterial fluid loss), circulates around and through the brain and spinal cord, then is constantly being lost (primarily to the venous and lymphatic systems). The dynamics of the CSF are greatly influenced, if not driven, by the fluid pressure of the CSF, but the origins and magnitude of the CSF fluid pressure are poorly understood. Earlier experimental and clinical studies documented how arterial pressure and the ventilatory cycle influence CSF pressure; recent studies, mainly using telemetric recordings, have shown that movement increases the magnitude and variation of CSF pressure. The present study was designed to explore the relative contributions of specific body movements to CSF fluid pressure; these movements include physiological (as occurs during ventilation), postural, or physical displacements (bending or locomotion). The study was performed by taking direct CSF pressure recordings from anesthetized and freely moving American alligators (*Alligator mississippiensis*). The results demonstrate that body movements have a significant influence on CSF pressure, often orders of magnitude beyond the influence of arterial pressure.

**Abstract:**

This study was undertaken to document how the cerebrospinal fluid (CSF) pressure varied during movements and physiological activities. Using surgically implanted pressure catheters; the CSF pressure was recorded from sub-adult American alligators (*Alligator mississippiensis*) under anesthesia and post-recovery. Pressures were recorded during physiological activities (the cardiac cycle; passive and active ventilation); manual manipulation of the anesthetized animals (foot sweeps; tail oscillations; and body bends); as well as voluntary movements post-recovery (changes in body tone; defensive strikes; and locomotion). The CSF pulsations associated with the cardiac cycle had the lowest mean amplitude (3.7 mm Hg); during active ventilation and defensive strikes; the alligators routinely generated CSF pressure spikes in excess of 100 mm Hg. The recorded CSF pressures appear to be caused by a variety of mechanisms including vascular pressure; fluid inertia; and possible physical displacement of the spinal cord. The results of the study suggest that any model of CSF dynamics or perfusion should incorporate the episodic high-pressure CSF pulsations associated with movement

## 1. Introduction

The cerebrospinal fluid (CSF) functions in the development [1] and metabolic maintenance [2] of the central nervous system. To fulfill either functional role, the CSF must circulate; studies have postulated that abnormal CSF circulation may cause a variety of neurological disorders, including syringomyelia [3], Alzheimer’s disease [4], and hydrocephalus [5,6,7]. Despite the recognized importance of the CSF fluid mechanics, little is known about the magnitude, or dynamic range, of “normal” CSF pressure [8].

Recordings of CSF pressure pulsations generally include a higher-frequency component corresponding to the cardiac cycle [9], and a lower-frequency component corresponding to the ventilatory cycle [10]. Changes in the internal pressure of the body cavities, the intrathoracic and intraperitoneal pressures, can also influence the CSF dynamics [11]; it is unclear how these cavity pressure changes relate to the cardiac and ventilatory influences [12]. Shifts in venous blood, with corresponding changes in venous blood pressure, can also alter the CSF dynamics [13,14].

The majority of the data on CSF fluid dynamics has been collected from animal or human subjects using magnetic resonance imaging (MRI) [15], or via surgical implantation of pressure sensors (under anesthesia). These two experimental techniques yield very different data. MRI sequences exclusively capture flow (not pressure); surgically implanted catheters have been described as the “gold standard” of CSF data [16], but they only collect pressure, not flow, data.

Recent telemetric studies of human movement [17,18,19] and rodent locomotion [20,21,22,23], have shown that movement results in an increase in CSF pressure. This increase in CSF pressure is thought to occur when the skeletal muscle contraction associated with motion/locomotion triggers an elevation of cardiac output which leads to increased cranial perfusion [24]. When kinematic features of locomotion have been quantified, there is a demonstrated relationship with CSF pressure [25] suggesting that impulses (shifts in fluid momentum) may explain some of the locomotor-based CSF patterns. Experimental stimulation of the myodural bridge, skeletal muscle fibers that insert onto the dural sheath [26], demonstrated that muscle contraction can directly influence CSF pressure. Accordingly, vertebrate motion can influence CSF dynamics in at least four ways; indirectly via the vascular response [24], by generating shifts in fluid momentum, by causing physical displacement of the spinal cord [27], and by skeletal muscles changing the dimensions of the CSF-filled spaces [28]. To date little is known about the relative contributions of these different influences, or how they vary, both interspecifically and among different forms of movement/locomotion.

The goal of the present study was to document the influence of movements, both fictive and natural, on CSF pressure, in the American alligator (*Alligator mississippiensis*). Recent studies have shown that *A. mississippiensis* has a well-developed myodural bridge, the contraction of which is capable of altering CSF pressure [29]; and that during locomotion the CSF pressure increases in a pattern suggestive of combined vascular and kinematic influences [30]. This contribution is intended to document the impacts on CSF pressure of other movements, and physiological behaviors.

## 2. Materials and Methods

### 2.1. Live Animals

Seven live sub-adult (142–165 cm total length, 8.8–14.7 kg mass) American alligators (*Alligator mississippiensis*) were obtained from the Louisiana Department of Wildlife and Fisheries. The animals were housed communally in a 29 m^2^ facility that featured three submerging ponds, natural light, and artificial lights on a 12:12 cycle. The facility was maintained at 30–33 °C, warm water rain showers were provided every 20 min, which helped maintain the facility at >75% relative humidity. The alligators were maintained on a diet of previously frozen adult rats. When the individual animals were removed from the enclosure, they were caught by noosing, then their jaws were taped using vinyl tape. The husbandry and use of the live alligators followed all applicable federal guidelines, and were approved by the IACUC of A.T. Still University (Protocol #221, approved March 2021).

### 2.2. Treadmill and Treadmill Training

A treadmill (270 cm long × 61 deep × 46 wide) was fabricated out of steel beams, lined with stainless steel sheeting, and fitted with a custom (538 cm long × 41 cm wide) tread (Walking Belts, LLC, Temecula, CA, USA). The two ends of the treadmill had sliding gates. Each alligator was trained twice weekly to locomote along the length of the treadmill, initially with the treadmill off, subsequently with it moving in the range of experimental velocities (0.1–0.25 m/s). All training sessions were recorded using two digital video cameras (Action camera, YI Technology, Shanghai, China), one providing a dorsal view and the other a head-on view. Once all of the alligators demonstrated stable locomotor patterns on the moving treadmill, the surgical experiments were initiated. One terminal surgical experiment was performed each week; the training schedule was maintained for all of the remaining animals.

### 2.3. Surgery and Data Collection

When the individual alligator was noosed for the surgical experiment it was induced to bite a bite pad, and the animal’s mouth was taped shut around the bite pad. Each individual alligator was placed on a stiff board (244 × 28 × 3.8 cm thick), which exceeded the maximum width and length of the alligators used for this study. Six 2.5 cm wide heavy duty straps (Northwest Tarp and Canvas; Bellingham, WA, USA) were used to secure the alligator to the board; the straps were tight enough to minimize movement of the animal but not tight enough to impede ventilation or circulation. With the alligator’s mouth held open by the bite pad, a laryngoscope was used to depress the gular valve [31] and expose the glottis. A cuffed endotracheal tube was inserted into the larynx and connected to a custom anesthesia system that included a ventilator pump (Harvard Apparatus, Holliston, MA, USA), Vaporstick anesthesia machine (Surgivet, San Clemente, CA, USA), isoflurane vaporizer (Surgivet), and Capnomac Ultima respiratory gas monitor (Datex-Engstrom, Tewksbury, MA, USA). The alligators were maintained on a steady ventilatory pattern of 5 breaths per minute each with a tidal volume of 500 mL. Anesthesia was accomplished using 5% isoflurane [32]. Two silver chloride surface cup electrodes (019-477200, GRASS, Natus Medical, Pleasanton, CA, USA), coated with a layer of conducting gel (Signagel, Parker Laboratories, Fairfield, NJ, USA) were placed on the lateral surface of the animal, on either side of the heart, and the EKG signal recorded (at 4 kHz) using a MiDAS (Xcitex Inc., Woburn, MA, USA) data acquisition system. Meloxicam (at 0.2 mg/kg) was administered into the left triceps to serve as an analgesic [33].

A surgical drill (MPS Powerforma, XOMED surgical products, Jacksonville, Fl, USA) was used to bore a 4.0 mm diameter hole through the dorsum of the alligator’s skull to expose the dura. A small incision was made in the dura to allow the passage of a pressure catheter. Surgical adhesive (Vetbond, 3M, Maplewood, MN, USA) was used to seal the dura around the catheter, then epoxy cement was added to fill the bored hole and secure the catheter to the skull. A fluid pressure transducer (APT300, Harvard Apparatus) was rested upon, and sutured to, the large osteoderm that covers the dorsum of the alligator’s neck. The pressure transducer, and the attached pressure catheter, were filled with a reptilian Ringers solution [34]. The lead coming off of the pressure transducer was sutured along the midline of the alligator to mid-body; neither the pressure catheter, transducer, nor the transducer’s lead, restricted the range of head or body movements of the alligator. The pressure transducer was coupled to strain gauge amplifier (P122, GRASS Instruments). The output from the amplifier was sampled at 4 kHz, simultaneously with the carbon dioxide concentration from the respiratory gas monitor, using the MiDas data acquisition system.

The experimental sequence for each alligator was identical and involved 10 stages or steps. Once the alligator reached a surgical plane of anesthesia: (1) the pressure catheter was implanted into the cranial sub-dural space and a series of baseline CSF, EKG, and (forced) ventilatory cycles were recorded. (2) the ventilator was turned off for 30 s intervals to document CSF pressure during apnea. (3) with the animal still under anesthesia it was tilted 30° head-up and 30° head-down (separately), each over a 90 s duration divided into 30 s baseline, 30 s tilt, then 30 s recovery. (4) the animal was allowed to recover from anesthesia, this is a slow (hours-long) process in alligators [32]; the animal was actively ventilated with oxygen during recovery, but no drugs were administered to accelerate recovery. (5) while the animal was still anesthetized, the straps restraining the tail were removed and the distal tip of the tail was manually wiggled from side to side. (6) the left hind limb was manually moved through a step-cycle. (7) the restraining straps on the trunk were removed, then the trunk and tail manually deflected to the left side. (8) once the animal was recovered enough to actively thrash against the restraining straps, the endotracheal tube and bite pad were removed (and the jaws again taped closed). The animal was then manually transferred to the adjacent treadmill and given time to adjust. (9) baseline recordings, now exclusively of CSF pressure, were taken while the animal was not moving on the treadmill. (10) once the alligator demonstrated coordinated voluntary movement, the treadmill was turned on and locomotor data collection commenced.

Typically 3–4 locomotor sequences were recorded from each alligator (see below). Immediately after the last locomotor sequence the treadmill was turned off and the alligator allowed to remain (stationary) on the treadmill while post-locomotor CSF pressures were recorded. The animal was then returned to the surgical board, re-anesthetized, then euthanized through cardiac excision and exsanguination [35]. Immediately after euthanizing the animal, the pressure catheter/transducer complex was calibrated.

The surgery and locomotor experiment were performed in the same small room which was maintained at >30 °C to prevent metabolic disruption to the alligators [36]. An LED flash was located adjacent to the treadmill to provide synchronization between the video and physiological data. A trigger switch controlled by one of the authors provide a second means of synchronizing the two data sets.

### 2.4. Data Analysis

The video records were analyzed using Kinovea (kinovea.org). The majority of the locomotor sequences recorded were not used for data analysis (although similar CSF patterns were evident). These sequences were excluded because the alligator failed to perform at least 3 complete footfall (step) sequences, collided with the end or sides of the treadmill, was physically contacted by either researcher, or exhibited locomotor kinematics (velocity, joint angles, etc.) that differed from those recorded during the final training sessions. Ultimately, 11 locomotor sequences were analyzed; these had a mean duration of 13.12 s (s.e. = 4.87). The physiological data was quantified with Midas, was exported to SpectraPlus (Pioneer Hill Software, Sequim, WA, USA) for Power Spectral Analysis, or was exported to EXCEL for statistical analysis.

The final data set consisted of an equal number of baseline, apnea, head-up and head-down rotations, manual tail oscillations, manual limb displacements, and manual body bends collected from all seven of the specimens. Baseline treadmill recordings were taken from all seven specimens. Nine defensive strikes were recorded (from four different specimens), and 11 locomotor sequences were quantified from 6 of the alligators (no more than 2 quantified sequences were examined from any one alligator).

## 3. Results

EKG traces were recorded once the animal was at a surgical plane of anesthesia; power spectral analysis of these records (Figure 1A) reveal a mean dominant frequency of 0.37 Hz, corresponding to a mean heart rate of 22 (s.d. = 5.0) beats per minute (bpm). This cardiac frequency had a 1:1 relationship with CSF pressure pulsations (Figure 1B). The CSF pressure increased following the QRS complex of the EKG, and decreased following the T wave of the EKG (Figure 1C). The cardiac-related pulsation in the CSF were best visualized during (induced) apnea, where they had a mean amplitude of 3.7 mm Hg (s.d. = 0.19).

Throughout anesthesia, and for most of anesthesia recovery, the alligator was connected to a mechanical ventilator set for 5 breaths per minute; accordingly, power spectral analysis of the exhalatory CO_2_ record yielded a mean dominant frequency of 0.8 Hz (Figure 2A). This passive ventilatory cycle produced a second stable pattern of CSF pressure pulsations (Figure 2B) with a mean amplitude of 9.2 mm Hg (s.d. = 0.32) and a frequency of 0.8 Hz. The CSF pressure pulsations linked to the passive ventilatory cycle were independent of those associated with the cardiac cycle; cardiac pulsation could occur at the peak of the ventilatory curves (further increasing CSF pressure) and the increase in CSF pressure at the onset of a ventilatory cycle could obscure the cardiac-related CSF pulsation (Figure 2).

While the animals were still anesthetized and connected to the ventilator, they were tilted to 30° head-up for 30 s, then (following a 30 s recovery period) tilted to 30° head-down for 30 s (Figure 3). The 30° head-up rotations caused a decrease in baseline CSF pressure of 16.8 mm Hg (s.d. = 2.4), and a decrease in the amplitude of the cardiac-related CSF pulsations of 2.6 (s.d. = 1.6) mm Hg. The 30° head-down rotations caused an increase in baseline CSF pressure of 18.6 mm Hg (s.d. = 1.23), and an increase in the amplitude of the cardiac-related CSF pulsations of 2.8 (s.d. = 1.4) mm Hg. Neither the heart-rate nor the cardiac-related CSF pressure pulses showed evidence of a compensatory response to these orthostatic gradients (Figure 3).

With the alligators under anesthesia, and the pelvic girdle and the body rostral to it mechanically restrained, the distal tip of the tail was wiggled manually to generate sinusoidal waves in the tail. Video analysis revealed that the sinusoidal tail waves had a mean frequency of 4.56 Hz (s.d. = 0.79), and that they propagated the length of the tail, which is roughly half the total length of the alligator. The CSF pressure traces show high-frequency pulses coincidental to the manual oscillations of the tail; these pulses are concurrent with the cardiac- and ventilatory-related pulses (Figure 4A,B). When examined using power spectral analysis, the CSF pulsations recorded during tail wiggling have a similar frequency to the tail oscillation (Figure 4C); and are significantly (paired *t*-test; *t* = 40.49, *p* < 0.00001, df = 13) higher in frequency than the heart rate (or cardiac-related CSF pulsation) recorded in the same trial. The individual CSF pulsations recorded during tail oscillation had a mean amplitude of 4.9 mm Hg (s.d. = 1.17).

Immediately following the tail oscillation trials, while the alligator was still under anesthesia, the left hind limb was manually moved through a step-cycle. That is, the pes was first slightly elevated and protracted, then moved caudally while maintaining a constant distance from the body. Every effort was taken to make the foot sweeps as smooth and repeatable as possible. Protraction of the hind limb caused a marked increase in cranial sub-dural CSF pressure (Figure 5), this pressure decreased with retraction of the pes (Figure 5). Manual foot sweeps resulted in CSF pressure pulsations with a mean amplitude of 9.4 mm Hg (s.d. = 2.08).

The restraining straps on the body were removed, so only those securing the head and neck remained. Then, using the pelvis and mid-tail at contact points, the body of the anesthetized alligator was manually deflected to one side (see Appendix A). Each deflection of the body caused an increased in the CSF pressure (Figure 6); the pulses of CSF pressure caused by deflection of the body had a mean amplitude of 35.6 mm Hg (s.d. = 11.28). Depending on where the bending occurred relative to the ventilatory cycle, the manual bending caused either a modest disruption of the normal airflow pattern, or had no obvious influence (Figure 6).

Following the body bends, the animal was allowed to recover from anesthesia. No drugs were administered to facilitate recovery, which is a long process in *Alligator* (mean = 215.5 min., s.d. = 74.6). During this recovery period data were recorded periodically. Near mid-recovery, the cardiac frequency was 0.39 Hz (s.d. = 0.07) corresponding to a heart rate of 23.5 bpm. The cardiac-related CSF pulsations recorded mid-recovery had amplitudes that were not significantly different from the baseline values (paired *t*-test; *t* = 0.189, *p* = 0.856, df = 13). During the recovery period the animal would start performing spontaneous breaths, herein defined as movement of the thoracic and/or abdominal body wall that altered the ventilatory airflow pattern. These breaths varied widely in duration (Figure 7), and may or may not have an initial inspiratory phase during which the CSF pressure decreased (Figure 7). Regardless of the duration, these breaths had a pronounced influence on the CSF pressure, often raising the cranial sub-dural CSF pressure to over 100 mm Hg (Figure 7).

When the animal was judged to have fully recovered from anesthesia, all leads except for the implanted pressure catheter, were removed and the animal was placed in a custom-built treadmill (which was not running). The animal was left undisturbed until it began independent sustained locomotion along the treadmill. CSF pulsations recorded during these non-locomotor periods (Figure 8A) had a mean frequency of 0.39 Hz (or 23.9 beats per minute, s.d. = 4.7) and mean amplitude of 3.8 mm Hg (s.d. = 0.19); these were not significantly different (paired *t*-test for frequency: *t* = 0.598, *p* = 0.572, df = 13; for amplitude: *t* = 0.989, *p* = 0.329, df = 34) from the cardiac-linked pulsations recorded while the animal was under anesthesia (Figure 8B). During this time, while not locomoting, the CSF pressure record and the movement of the animal’s body were suggestive of active breathing (Figure 9). Since the exhalatory CO_2_ was no longer being monitored, the designation of active breathing is only provisional. During this time, there were also marked changes in baseline CSF pressure (mean 12 mm Hg, s.d. = 15.25) which seemed to be related to postural shifts and associated changes in body tone (Figure 10). The animals would extend their limbs lifting their body off the treadmill surface (lowering baseline CSF, Figure 10A) or would sprawl onto the surface of the treadmill, compressing their body cavities with their body mass (raising baseline CSF, Figure 10A). The term “body tone” is used to convey the changes in the width of the alligator’s trunk that were evident in the video record whenever the animal would switch between resting on the substrate or supporting itself on its limbs.

Once fully recovered the alligators would frequently exhibit defensive behaviors, particularly hissing, tail whips, and defensive strikes. During the “typical” defensive strike the body is bent sharply to one side as both the tail and head are elevated and rapidly turned in the same direction (Figure 11A,B). These defensive strikes produced increases in CSF pressure that had a mean of 76.7 mm Hg (s.d. = 17.6). Most commonly, this increased pressure was a short-duration “spike” (Figure 11C), but this was quite variable depending on the velocity of the strike, the extent of the bend, and the relative change in body inflation (Figure 11D). One series of three consecutive defensive strikes was recorded (Figure 11E); the elevated pressure between these three strikes may reflect the continuous hissing the animal was performing.

When the animal was locomoting over the treadmill, whether or not the treadmill was running, the CSF pressure underwent low frequency (mean 0.54 Hz, s.d. = 0.09) sinusoidal changes (Figure 12A) with mean amplitudes of 59.5 mm Hg (s.d. = 12.19). In all of the locomotor trials, there was a close temporal congruence between the lateral oscillations of the alligator’s head during locomotion, and the sinusoidal pattern of CSF pulsations (Figure 12B). Regression analysis revealed a significant (F = 51.26, *p* = 0.00005, df = 1, R = 0.92) relationship between increasing oscillatory frequency of the head and the frequency of the CSF pulsations. In addition to these sinusoidal pulses, locomotor activity caused a general increase in CSF pressure with a mean pressure increase of approximately 20 mm Hg (Figure 12A,C).

The cardiac-related CSF pressure pulsations were the smallest pulsations recorded that could be clearly linked to a physiological feature or activity (Table 1). Some of the relative influences are unexpected; “sweeping” the hindlimb through a step cycle has almost twice the impact on CSF pressure as does oscillating the entire tail (Table 1). The largest influences on CSF pressure amplitudes are those that involve the entire body, rapid movement, or internal force generation (Table 1).

## 4. Discussion

The present study was undertaken to document the influence of movement, including locomotion, on CSF pressure. To minimize variation, a restricted size range of *Alligator mississippiensis* was used in this study, all seven animals were trained to locomote on the same treadmill, the same surgical approach was used on each animal, and the sequence of data collection was held constant. The CSF pressure pulsations linked to the cardiac cycle had the lowest amplitudes recorded. Changes in body tone, ventilatory movements, and passive (under anesthesia) or active (after recovery) movements all generated shifts in the CSF pressure baseline, or CSF pulse amplitudes, that were larger, in some cases more than an order of magnitude larger, than those attributed to the cardiac cycle.

This study used surgical implantation of a pressure catheter to record CSF pressure in the subdural space; this is a well-established technique [37] that is still considered the gold-standard for recording CSF pressure [16,38]. In humans, the catheter is normally positioned into the lateral or 3rd ventricle, and can also serve as a CSF drain or a means of intrathecal drug administration [39,40,41]. The EVD (External Ventricular Drain) approach can lead to contamination and infection [42,43] which were not an issue in the present study, as well as hemorrhage [44], and unintended positioning of the catheter [45]. The present study used a variation of this approach to catheter placement, in part to ensure that the catheter would have minimal impact on the CSF dynamics. The surgical bore hole in the skull was only slightly larger than the pressure catheter, and the incision in the dura was no larger than the catheter. The catheter was positioned as superficially as possible, the goal was to keep the open port of the catheter at the level of the deep surface of the dura. The integrity of the dura was ensured by sealing the catheter to the dura using surgical adhesive; no leakage was observed during any of these preparations. The dorsum of the alligator skull is considerably thicker than the human skull. The deep surgical bore hole used to place the catheter was filled with epoxy glue once initial CSF pressure recordings were obtained. This further prevented CSF leakage and created a robust anchoring point for the catheter. The main limitations of the present study were: (1) the implanted catheters could not record CSF flow rates; (2) alligators have variation in their CSF fluid dynamics [46] similar to humans [8]; and (3) some of the movements analyzed in this study were performed manually, which produced additional variation.

The present study was designed to explore the influence of movement on CSF pressure, not the CSF fluid dynamics of *Alligator mississippiensis*. Accordingly, our emphasis is not on the absolute value of the CSF pressure (though those values are all provided) since this would change over the enormous size range of *A. mississippiensis* [47] and would be greatly influenced by specifics of the alligator’s neuroanatomy [48]. Instead, we used as a relative scale the CSF pressure amplitude caused by each cardiac cycle (Table 1); these cardiac-related pressure pulses have been found in the CSF of every vertebrate examined [15,49], and can be directly visualized in the CSF pressure traces presented herein.

The heart rate of the seven *Alligator mississippiensis* was recorded while the animals were under anesthesia. The pooled heart rates: immediately post-surgery (mean = 22 bpm), mid-recovery (mean = 23.5 bpm), and immediately upon full recovery (mean 23.9 bpm); did not significantly change over the course of the experiment and are comparable with earlier studies of anesthesia and heart rate in *Alligator* [50]. Like the frequency, the amplitude of the cardiac-related CSF pulsations did not vary significantly through the course of the experiments (mean of 3.7 mm Hg, Figure 8). Because of the consistency in the frequency and amplitude of the cardiac-related CSF pulsations, they could be reliably identified in the recorded data traces.

The change in CSF pressure observed when the animal is exposed to an orthostatic gradient (in this study the orthostatic gradient was generated by rotating the alligator into a head-up or head-down position) was used, in part, to check the integrity of the experimental preparation. Every animal exhibited a similar magnitude of CSF pressure change upon rotation (Figure 3), these magnitudes were similar to what was reported in previous studies [51,52]. During head-down rotations the cardiac-related CSF pulsations increased in amplitude, then decreased in amplitude during head-up rotations; presumably this reflects the change in cardiac output caused by shifts in venous return.

The ventilatory-related CSF pressure pulsations, though easily identified (Figure 2, Figure 7 and Figure 8), were highly variable. The majority of this variation is due to two factors. Firstly, the anesthesia protocol developed in the lab (which produced the traces in Figure 2) was designed to safely achieve and maintain a surgical plane of isoflurane anesthesia [32,53], it was not designed to replicate the mechanics of active breathing in *A. mississippiensis*. Secondly, this species has some of the most variable ventilatory mechanics of any vertebrate. Crocodilians have the ability to switch between thoracic, abdominal, and pelvic aspiration [54,55], active control over the glottal aperture [56], and a high tolerance for hypoxia [57] which manifests as intermittent breathing [58]. The trace shown in Figure 7B was an audible cough; the traces shown in Figure 9 both appear to be “deep” breaths, one of which (Figure 9B) was followed by breath holding while the other (Figure 9A) was followed by two shallow breaths. A full examination of the relationship between the diversity of ventilatory mechanics and CSF pressure in crocodilians is beyond the scope of the present paper; this contribution was intended to document that the ventilatory mechanics of crocodilians were capable of producing CSF pressure pulses with amplitudes far greater than those associated with the cardiac cycle (Table 1).

Since *A. mississippiensis* has a functional diaphragm [59], the animal can actively regulate the intraperitoneal or intrathoracic pressures, and in that way cause sustained shifts (positive or negative) to the baseline CSF pressure values (Figure 10). Similar baseline shifts could also be produced “passively”; some of these alligators had a mass greater than 14.5 kg so when the animal sprawled (resting on the substrate without limb support) the body cavities were under different compression than when the animal was using its limbs for support. These changes in compression were evidenced on the video records as changes in the width of the trunk of the alligator. These shifts in the CSF pressure baseline had a mean magnitude of 12 mm Hg, which is over 3 times the mean amplitude of a cardiac-related CSF pressure pulsation (Table 1).

The other changes in CSF pressure documented in this study are likely the product of changes in fluid momentum of the CSF (an impulse), a mechanical impingement on the sub-dural space (where the CSF is located in *Alligator mississippiensis*) by the vertebrae, physical displacement of the spinal cord, or a mechanical impingement on the peritoneal or thoracic cavity. These four causal mechanisms may not be discrete, nor need they be exclusive.

When the hind limb was manipulated through a step-cycle no movement was observed in the rest of the body (which was still mechanically restrained). Accordingly, the spikes in CSF that were produced by moving the limb (Figure 5) are difficult to interpret as a CSF impulse. It is possible that a tendon or muscle (such as caudofemoralis [60]), which attached to the vertebrae then spanned the acetabular/femoral joint before attaching distally on the hind limb, could cause displacement of the dura during manual manipulation of the limb. Nevertheless, herein we hypothesis that the CSF pressure pulsations recorded during manual manipulation of the hind limb were caused by the hind limb compressing the peritoneal cavity during protraction, then allowing decompression of that space during retraction. In any case, this manipulation produced a mean increase in CSF pressure of 9.4 mm Hg or some 2.5 times the CSF pulse amplitude attributable to the cardiac cycle (Table 1).

Lateral deflection of the body of *Alligator mississippiensis*, which was done manually (Figure 6) and as part of defensive strikes (Figure 11), could impinge on the sub-dural space through either vertebral or spinal cord displacement; alternatively, deflection of the body could influence the CSF pressure by altering the intrathoracic and/or intraperitoneal pressures. The latter mechanism seems less likely since we recorded clear spikes in CSF pressure during manual bending which were not correlated with any change in the exhalatory CO_2_ pattern (Figure 6). The manual bends performed on the anesthetized animals were slower and involved less deflection than the defensive strikes performed by the alligators after recovery. The CSF pressure increase associated with the manual bending (mean = 35.6 mm Hg) was just under half the value recorded during defensive strikes (76.7 mm Hg). The CSF pressure increases produced during these lateral deflections were up to 20x greater than the amplitude of the cardiac-related CSF pulsations (Table 1).

We hypothesize that the pressure pulsations recorded during manual oscillation of the tail and during locomotion were caused by impulses (changes in fluid momentum) acting on the CSF. These two oscillatory movements were quite different (as evident in the Appendix A); the tail oscillations were of higher frequency and amplitude than the body and head oscillations that occurred during locomotion. In both cases, there was a match between the frequency of oscillation and the frequency of the CSF pulsations (Figure 4 and Figure 12). Yet, there was an enormous difference in the amplitude of the associated CSF pulsations; with tail oscillations producing mean CSF amplitudes of 4.9 mm Hg while locomotion produced mean CSF amplitudes of 59.5 mm Hg (Table 1).

The marked difference in the amplitudes of these two oscillatory CSF pressure pulses may reflect the unusual anatomy of the tail of *Alligator mississippiensis*. Though the tail makes up roughly half the total body length of the alligator [61], it holds only a small fraction of the CSF in part due the absence of a caudal cistern (a large lumbar cistern is present) [62]. This smaller CSF volume means oscillations of the tail would produce less fluid momentum, and the resulting pulses would be damped by passing through the lumbar cistern; in contrast, the oscillations of the body and head during locomotion produce CSF pulses with high momentum which propagate to the head without passing through a dampening cistern.

Another key difference between manual tail oscillations and the oscillations of the body and head during locomotion, is the potential influence of the myodural bridge. The myodural bridge is a specialization of the suboccipital muscles in which muscle fibers insert onto the dura adjacent to the first cervical vertebra (the atlas) [63]. Though other functions have been proposed [64] there is a consensus that the myodural bridge could influence CSF dynamics during turning (oscillating) the head. Experimental work has shown that stimulation of the myodural bridge of *Alligator mississippiensis* does alter CSF pressure [29], though the magnitude of this alteration under natural conditions is unknown.

Post-surgery, while the animals were still under anesthesia, cardiac and (passive) ventilatory-associated pulses in the CSF pressure were recorded, which, at 3.7 and 9.2 mm Hg, respectively, were among the smallest influences recorded. Once the animals were recovered from anesthesia, as evidenced in part from their independent and sustained movements, statistically similar cardiac-related CSF pressure pulses were recorded (Figure 8), but now body tone/movement, active ventilation, defensive strikes, and locomotion were all found to produce pulses in CSF pressure which were substantially greater (Table 1) than those recorded while the animal was anesthetized. This is a critical issue in that the majority of the literature on CSF dynamics was obtained in one of three ways: (1) experimental studies using anesthetized animal subjects, (2) experimental studies using anesthetized, and often clinically compromised, human patients; and (3) studies of CSF flow using MRI analysis which requires the subject (human or animal) to remain relatively still. The overwhelming prevalence of these three data sources may skew our perception of the CSF fluid dynamics as being a rather stable low-pressure system, and may help explain why there is so much uncertainty or variation regarding what is considered “normal” CSF pressure [65,66].

Evidence for a more dynamic system, capable of coping with high-pressure pulses of CSF pressure, can be found in the literature from studies that recorded from human subjects which had recovered from anesthesia. A study of coughing [67] noted that during coughing, “amplitudes above 100 mm Hg could be produced by most subjects”. The Valsalva manoeuver increased CSF pressure by 10.5 mm Hg [68], and raising intraabdominal pressure to (16 mm Hg) caused a 150% increase in CSF pressure [69].

Telemetric studies of human CSF pressure typically publish data records over longer intervals which obscure the cardiac pulsations, but do show clear variation and marked increases in CSF pressure often attributed to movement or postural changes [70,71,72]. The closest parallel to the present study may come from previous investigations in which the CSF pressure of rats was continuously recorded after the animals had recovered from anesthesia. Starcevic et al. [22] noted that much wider variation was recorded in studies of conscious active animals than in studies using only anesthetized animals. The study of Starcevic et al. on freely-moving rats [22] found a mean CSF pressure range of 11.7, and a maximum range of 21.5 mm Hg, which correspond to roughly 8–14× the cardiac pressure pulsation (~1.5 mm Hg) in rats. Other studies of csf pressure in freely-moving rats [23] or mice [20,21] also noted short-term pressure spikes that were roughly 8× the mean baseline, and a great deal of CSF pressure variation. A previous study of CSF dynamics in freely moving snakes found similar oscillatory CSF pressure pulses which greatly exceeded the cardiac-related CSF amplitudes [25].

While moving on the treadmill the alligators used variable speeds and a mix of “low walk” and “high walk” postures [73], which were associated with limited vertebral and cephalic deflections [74]. The treadmill locomotion produced a two-part response in the csf pressure (Figure 12); one characterized by high-amplitude sinusoidal oscillations, and the other by a gradual increase in the baseline csf pressure. An earlier analysis of the locomotor influence on csf pressure [30] attributed the sinusoidal oscillations to fluid momentum, akin to the earlier study of head-turning in humans [75] and the baseline shift to an increase in cerebral perfusion similar to what has been shown in rodents [24].

The consistent findings of the present study, and the general similarity between these results and the literature on telemetrized CSF recordings from humans and other mammals, suggests that the dynamics of the CSF in freely-moving organisms have likely been under-estimated.

## 5. Conclusions

This study was performed to document the influence specific body movements have on the cerebrospinal fluid (CSF) pressure in the American alligator. Surgically implanted pressure catheters were used to record the CSF pressure; simultaneous recordings of the EKG, exhalatory gases, and video recordings of the animals, established a quantitative relationship between movement and changes in CSF pressure. The cardiac related pulsations had the lowest amplitude of all the CSF pressures. Physical displacement of the body, as during locomotion or defensive strikes, produced pulses of CSF pressure orders of magnitude greater than the cardiac-related pulsation; the more of the body that moved, and the faster it moved, the greater the recorded CSF pressures. Physiological movements, as during active ventilation, or active changes in body cavity pressure, were quite variable (producing both positive and negative CSF pressure changes); coughing produced some of the largest CSF pulse amplitudes recorded. The link between body movement and CSF pressure/flow warrants additional analyses as we work to understand the role of CSF dynamics in maintaining the health of the brain and spinal cord.

## Figures and Tables

**Figure 1 biology-11-01702-f001:**
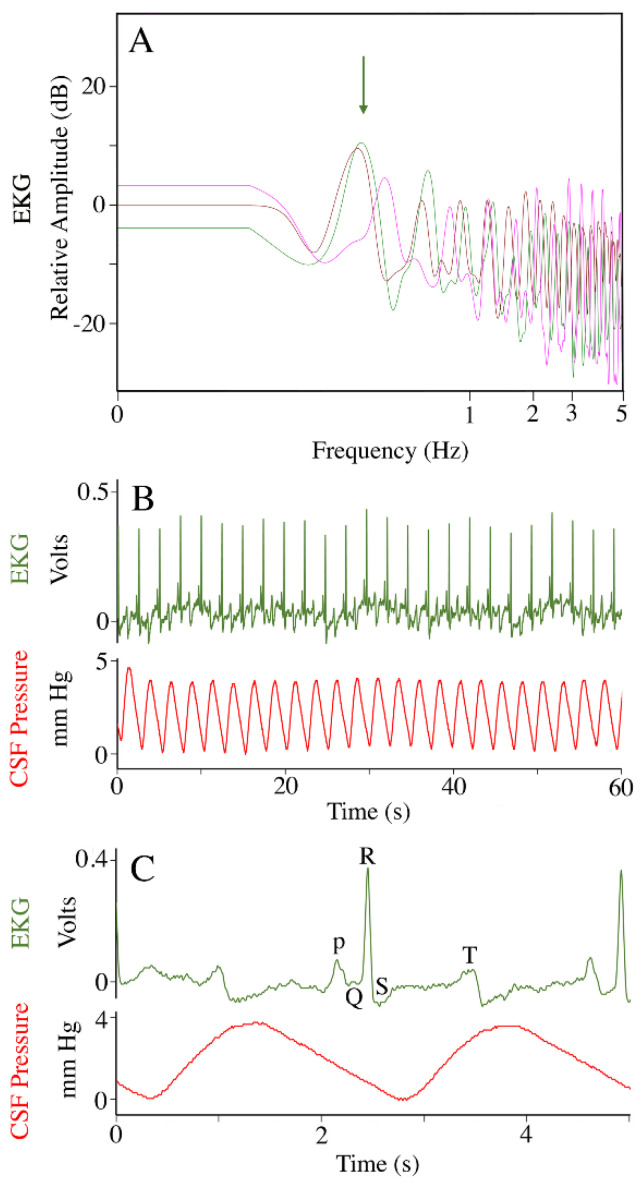
The EKG and cardiac-related CSF pressure pulsations. These sequences were recorded during a period of induced apnea, so no ventilatory influence is evident. (**A**) Power spectral analyses of the EKG traces from three individuals of *Alligator mississippiensis* (individuals’ traces are color-coded), all have a dominant frequency near 0.4 Hz (vertical arrow) corresponding to a heart rate of 24 beats per minute. (**B**) Simultaneously recorded EKG (dark green) and CSF pressure (red) tracings, showing the clear identification of the cardiac-related CSF pressure pulses. (**C**) simultaneously recorded EKG (dark green) and CSF pressure (red) tracings; the CSF pressure rises after the ventricular expulsion of the arterial blood (the QRS phase of the cardiac cycle). c—individual cardiac-related pulsation.

**Figure 2 biology-11-01702-f002:**
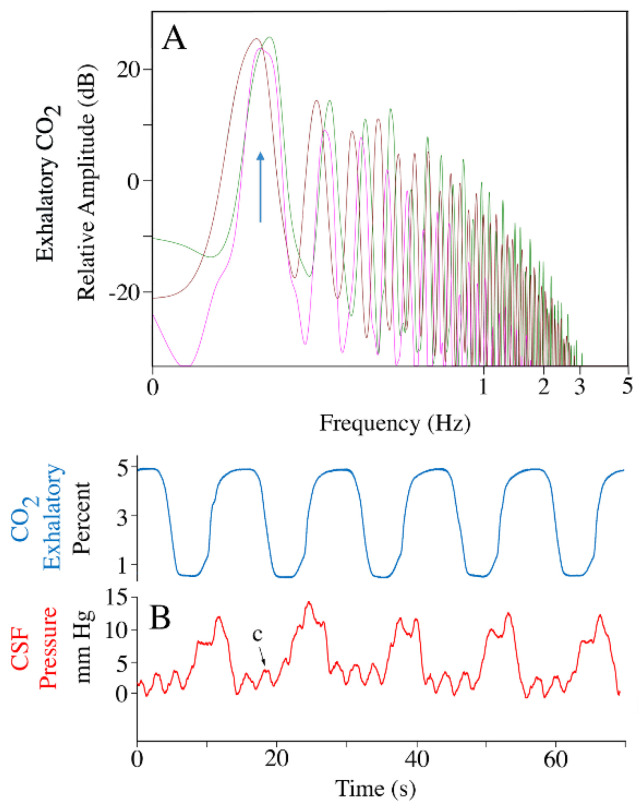
Exhalatory CO_2_ and ventilatory-related csf pressure pulsations. (**A**) Power spectral analyses of the exhalatory CO_2_ traces from three individuals of *Alligator mississippiensis* (individuals’ traces are color-coded), all have a dominant frequency near 0.08 Hz (vertical arrow) corresponding to a ventilatory rate of 5 breaths per minute. (**B**) Simultaneously recorded exhalatory CO_2_ (blue) and CSF pressure (red) tracings, showing the distinction between the ventilatory- and cardiac-related CSF pressure pulses. c—individual cardiac-related pulsation.

**Figure 3 biology-11-01702-f003:**
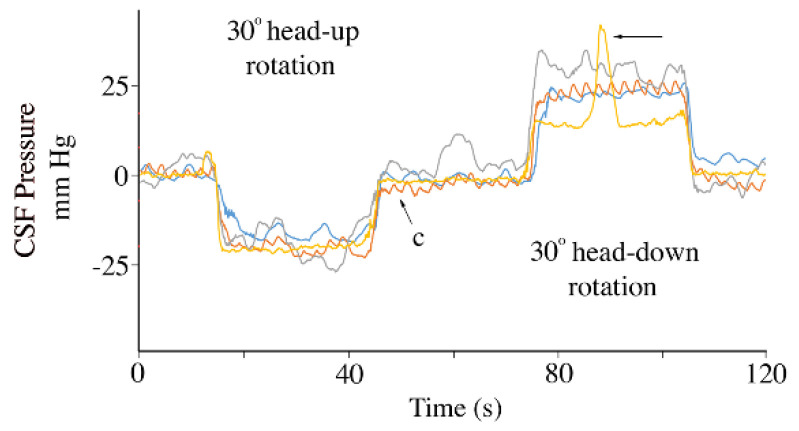
Influence of orthostatic gradients on the cerebrospinal fluid pressure. Traces from four individuals of *Alligator mississippiensis* (individuals’ traces are color coded), all showing similar scales of decreased CSF pressure during head-up rotations and increased CSF pressure during head-down rotations. One of the alligators coughed during the head-up rotation (arrow). c—individual cardiac-related pulsation.

**Figure 4 biology-11-01702-f004:**
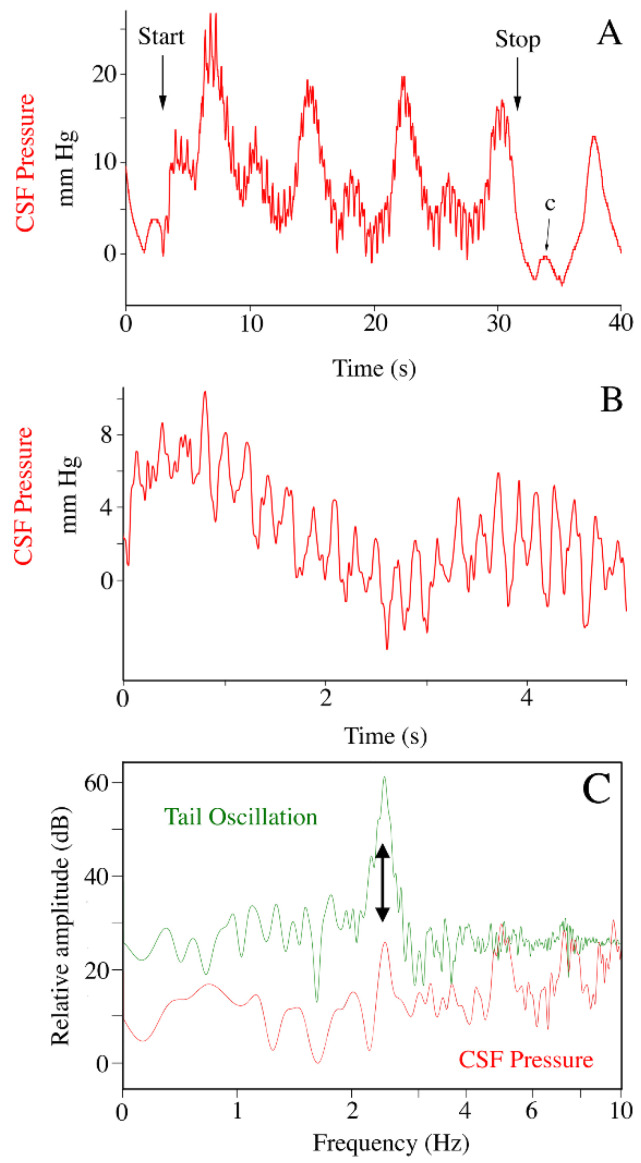
The influence of tail oscillations on CSF pressure. (**A**) The manual oscillation of the tail (vertical arrows) produced a distinct pattern of CSF pulsations (red) which obscured the cardiac-related pulsations, but were additive with the ventilatory-related pulsations. (**B**) Expanded time scale showing the higher frequency sinusoidal CSF pressure waves (red) recorded during tail oscillation. (**C**) Power spectral analyses of displacement of the tail (green) and the corresponding CSF pressure (red) recordings; note the overlap in the dominant frequencies (arrow), which (near 2.5 Hz) are well above the cardiac-related frequencies. c—individual cardiac-related pulsation.

**Figure 5 biology-11-01702-f005:**
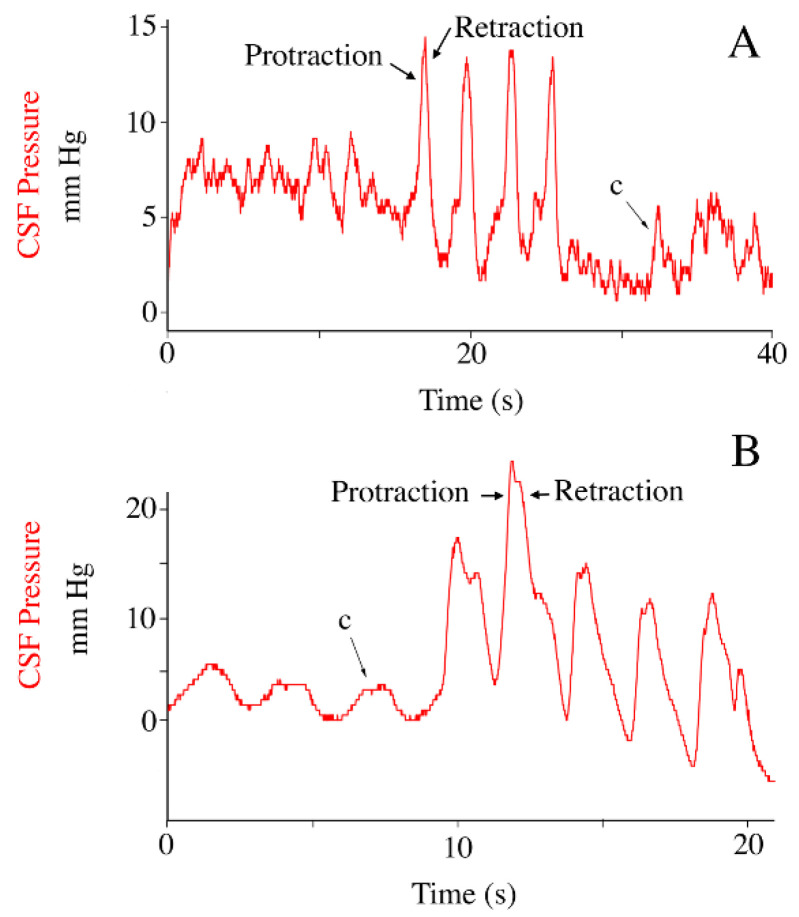
The impact of hind limb displacement on CSF pressure. Traces from two individuals of *Alligator mississippiensis* (**A**,**B**) showing the similar patterns of CSF pulsations (red) recorded during movement of the pes. Protraction of the pes caused an increase in CSF pressure, which decreased during retraction of the pes. c—individual cardiac-related pulsation.

**Figure 6 biology-11-01702-f006:**
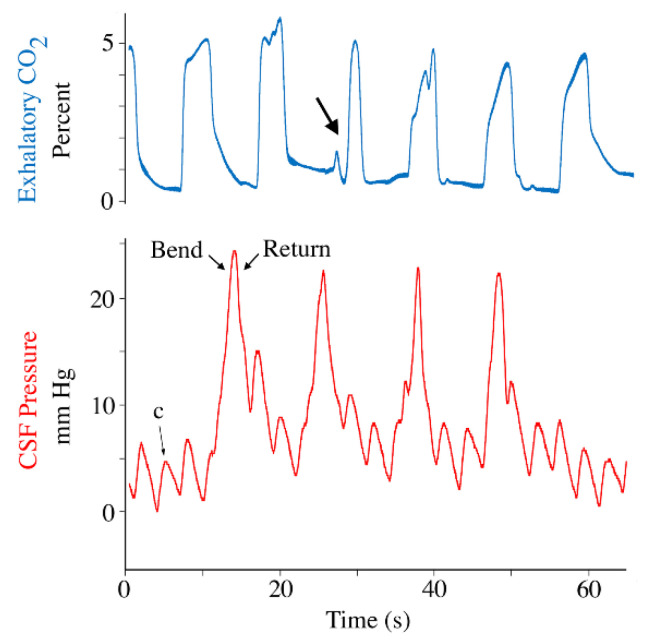
The influence of manual deflection of the body on CSF pressure. When the anesthetized alligator was bent to one side (see Appendix A) there was a marked increase in CSF pressure (red). The exhalatory airflow (blue) was generally not changed during the bends, but some of the cycles were shortened and smaller spikes of airflow were observed (arrow). c—individual cardiac-related pulsation.

**Figure 7 biology-11-01702-f007:**
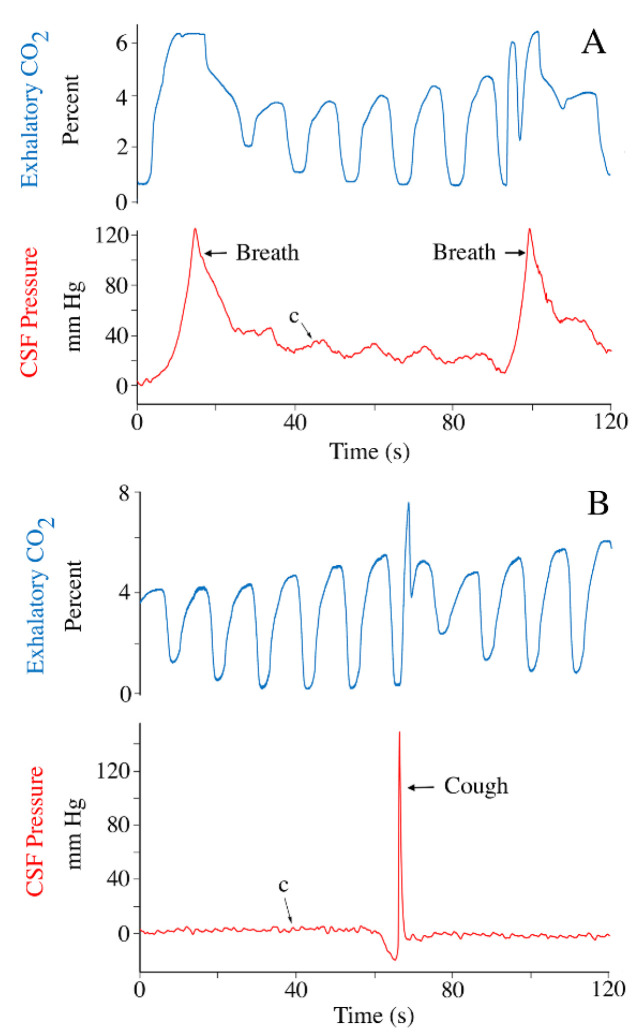
Records of active breathing during recovery from anesthesia. One of the alligators (**A**) took two deep breaths, while the other alligator (**B**) coughed. All of these active breaths changed both the CSF pressure (red) and the pattern of exhalatory CO_2_ (blue). c—individual cardiac-related pulsation.

**Figure 8 biology-11-01702-f008:**
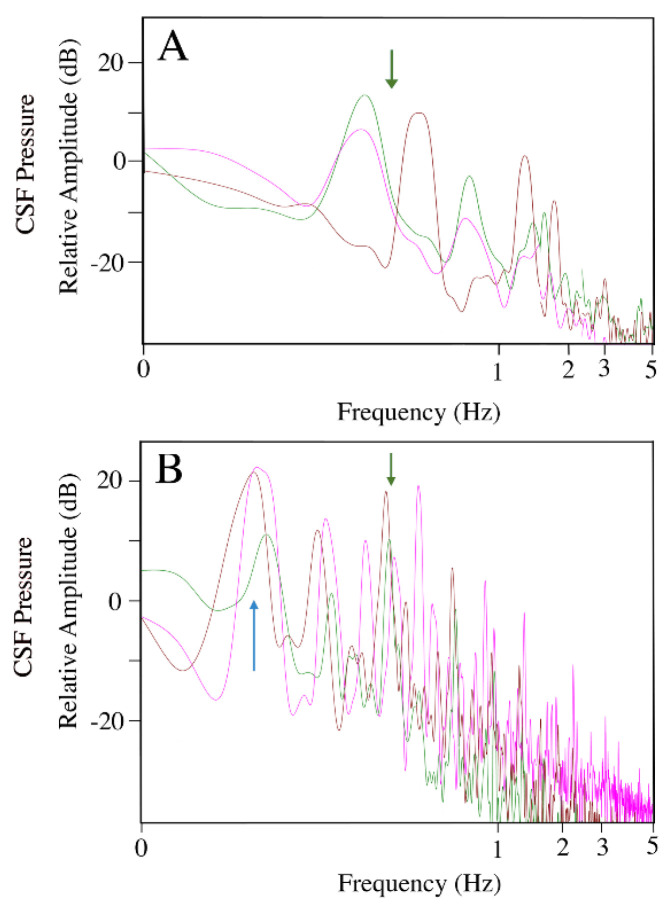
Consistency of the cardiac-related CSF pulsations. Power spectral analyses of three different specimens of *Alligator mississippiensi* (individuals’ traces are color coded); data were taken post-recovery during the locomotor trials (**A**) and immediately pre-surgery (**B**). The pre-surgery records included a ventilatory component (blue arrow). While there was some variation in heart rate, all of the traces had a clear cardiac-related frequency (green arrow) which was distinct from all of the other frequencies recorded during the experiment.

**Figure 9 biology-11-01702-f009:**
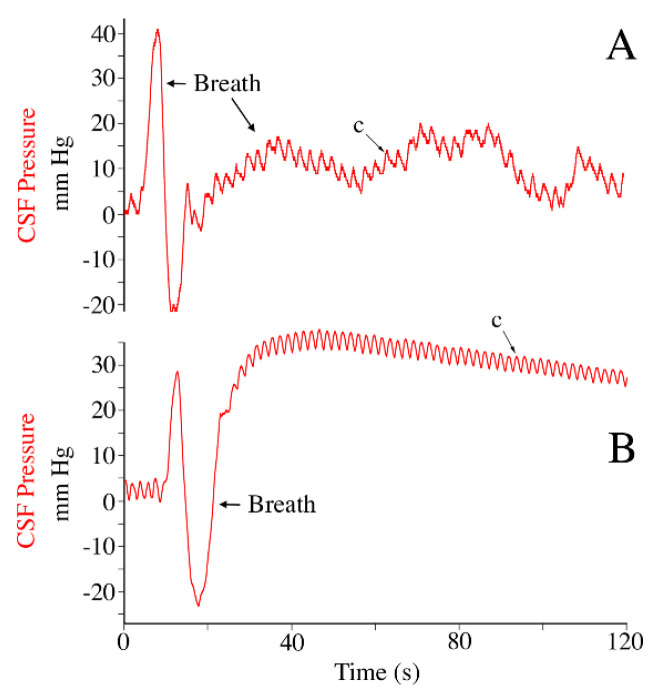
Impact of active breathing on CSF pressure. Pressure traces from two specimens of *Alligator mississippiensis* recorded during the locomotor trials (post-recovery) show the marked change in CSF pressure (red) during active breathing. One alligator (**A**) performed a deep active breath followed by two shallow breaths (the latter similar to those induced by the ventilator). The second alligator (**B**) performed a deep active breath then entered a period of apnea with sustained elevated CSF pressure. c—individual cardiac-related pulsation.

**Figure 10 biology-11-01702-f010:**
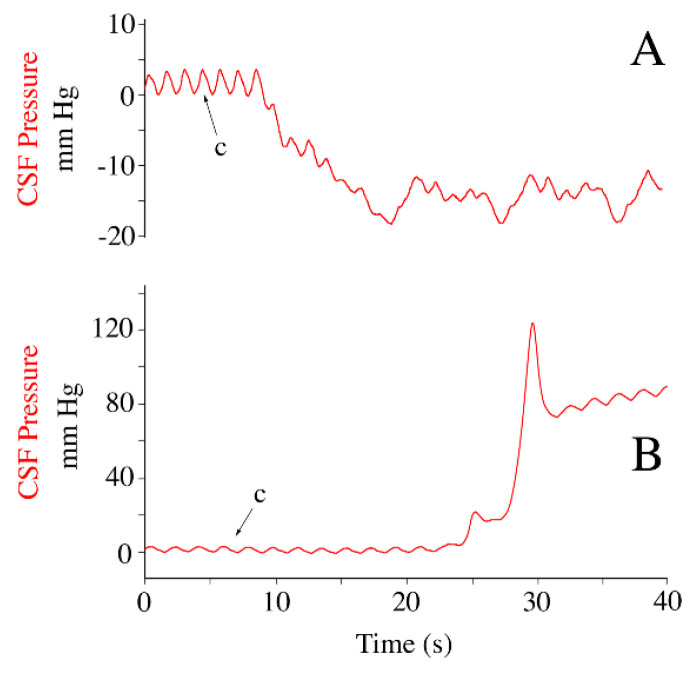
The influence of body tone on CSF pressure. During the locomotor trials (post-recovery) the alligators could shift their body tone, most commonly by lowering their body onto, or elevating it off of, the substrate. (**A**) a decrease in baseline CSF pressure (red) occurring as the alligator elevated up off of the substrate (and decreased the compression on its trunk). (**B**) a sharp increase in baseline CSF pressure (red) that occurred when the alligator sprawled out onto the substrate (and increased the compression on its trunk). Note that these changes in baseline CSF pressure were achieved without body movements suggestive of ventilation. c—individual cardiac-related pulsation.

**Figure 11 biology-11-01702-f011:**
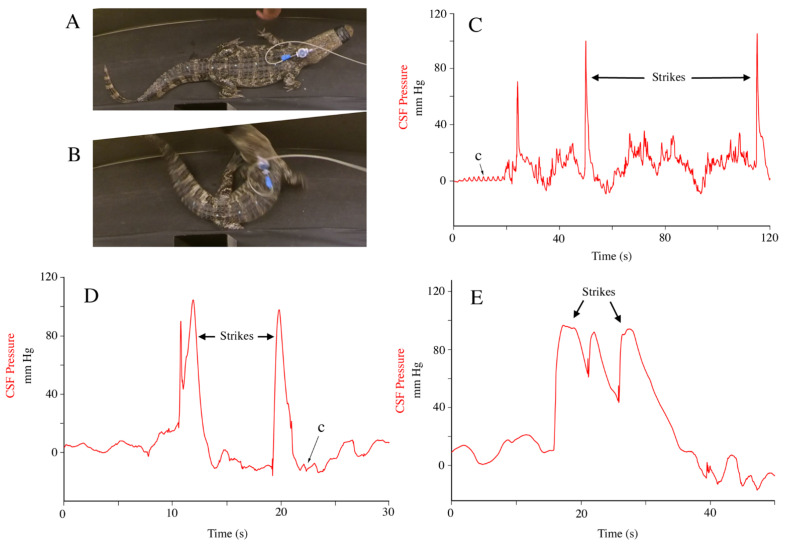
Defensive strikes of *Alligator mississippiensis* produced increases in CSF pressure. (**A**,**B**) isolated frames of a 150 cm *A. mississippiensis* moving both the tail and head to produce a sharp bend in the body while performing a defensive strike; there is 0.18 s between the two frames. (**C**–**E**) are records of defensive strikes performed by separate animals; in each case the elevation of CSF fluid pressure (red) during the strike is distinct from the cardiac-related pulsations. c—individual cardiac-related pulsation.

**Figure 12 biology-11-01702-f012:**
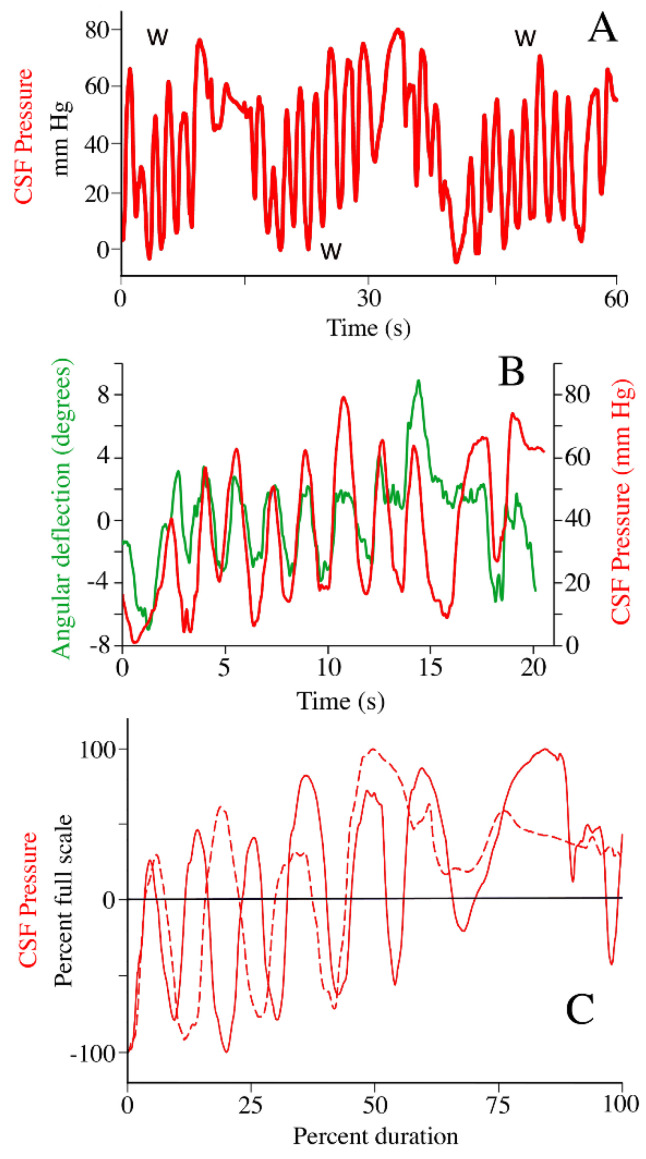
The influence of locomotion on CSF pressure in *Alligator mississippiensis*. Unrestrained animals were previously trained to walk on the treadmill; these data tracings were all taken after the animal was recovered enough from surgery to perform sustained locomotion. (**A**) CSF pressure traces (red) from an unrestrained alligator locomoting on a treadmill, the periods of sustained locomotion (denoted by W) are characterized by large amplitude sinusoidal waves of CSF pressure. (**B**) close temporal congruence between angular displacement of the head (green) and CSF pressure (red) during locomotion. (**C**) two locomotor CSF traces graphed as percent CSF pressure; note that the CSF pressure increases over the course of the locomotor sequence so that the pressures are above average (black horizontal line) at the end of the sequence.

**Table 1 biology-11-01702-t001:** Influences on Cerebrospinal fluid pressure in *Alligator mississippiensis*. The mean pulse amplitudes recorded during each activity, or physiological event, are given both in mm Hg, and in relation to the mean cardiac pulsation.

	Mean (mm Hg)	Cardiac Units
Resting cardiac	3.7	1.0
Passive ventilatory	9.2	2.5
30 head-up tilt	−16.8	−4.5
30 head-down tilt	18.6	5.0
Tail oscillation	4.9	1.3
Foot sweeps	9.4	2.5
Manual body bends	35.6	9.6
Body tone	12	3.2
Active ventilation	43.7	11.8
Defensive strikes	76.7	20.7
locomotion	59.5	16.1

## Data Availability

The data are available through reasonable request to the corresponding author.

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
