# Peer review of "The Influence of Movement on the Cerebrospinal Fluid Pressure of the American Alligator (Alligator mississippiensis)"

_biology, 2022, doi:10.3390/biology11121702_

Round 1

Reviewer 1 Report

Overall I very much enjoyed this manuscript and recommend that it be accepted after minor revisions.  The data presented is clear and the interpretations are sound.

My biggest overall complaint with the current manuscript is that it presents the notion that body movements contribute to ICP as a novel concept, particularly in the Simple Summary, when this is not at all a novel concept.  The authors note similar work in other animals later in the paper.

My second biggest complaint (which is somewhat intertwined with the first) relates to the fact that ICP studies in ambulatory humans are not cited and the authors, particularly in paragraph #3 of page 2 (Intro) and in the Discussion section, suggest to the readers that such data does not exist.  As a practicing pediatric neurosurgeon I not uncommonly do ambulatory ICP monitoring with patients, particularly those patients suspected of having idiopathic intracranial hypertension that do not have papilledema on fundoscopic exam.  Based on this personal experience I very much believe all the data presented as it agrees with clinical experience in humans. So, I think the authors need to include some more references about the human data ICP data that has been published.  There are many examples but here are a couple that come to mind:

1) Telemetric ICP monitoring systems exist, see:

Tschan CA, Velazquez Sanchez VF, Heckelmann M, Antes S. Home telemonitoring of intracranial pressure. Acta Neurochir (Wien). 2019 Aug;161(8):1605-1617. doi: 10.1007/s00701-019-03959-5. Epub 2019 Jun 5. PMID: 31168730

2) There are numerous studies in humans, including those without CSF disorders, looking at ICP recordings in mobile individuals.  Here is one that involved ICP monitoring during a mountain climbing expedition as an extreme example:

Wilson MH, and Milledge J. (2008). Direct measurement of intracranial pressure at high altitude and correlation of ventricular size with acute mountain sickness: Brian Cummins’ results from the 1985 Kishtwar expedition. Neurosurgery 63:970–974; discussion 974–975.

My third thought when reading this as a non-alligator expert was that it would have been nice to have a little primer on unique anatomic features of the alligator relevant to this study in the introduction.  For example the lack of a caudal cistern and presence of a prominent lumbar cistern in the alligator (mentioned lines 477-479 in Discussion) would have been helpful to know about prior to interpreting the results.

Highly specific notes:

- Line 40: It pains me that hydrocephalus, the canonical abnormal CSF disorder is not mentioned.  I would personally recommend adding it to the list of syringomyelia and Alzheimer's

- Line 97: "fit" should be "fitted"

- Figure 10: The figure legend is inadequate.  The body movements that produced the A and B recordings should be described.

- The discussion section gets repetitive.  In particular please note the similarity in content between lines 412-413 & lines 425-427

Author Response

Overall I very much enjoyed this manuscript and recommend that it be accepted after minor revisions.  The data presented is clear and the interpretations are sound.

 I thank you for your kind comment, and truly value your perspective on the manuscript.

My biggest overall complaint with the current manuscript is that it presents the notion that body movements contribute to ICP as a novel concept, particularly in the Simple Summary, when this is not at all a novel concept.  The authors note similar work in other animals later in the paper.

 I want to apologize to the Reviewer; it was never my intention to imply that this was the first study to explore movement contributions. I have reworded the Simple Summary, and try to clarify elsewhere in the manuscript, that the significance of movement has been previously recognized. The point I wanted to make was not that looking at movement was novel, but that this study was designed to apply a novel level of resolution to movement....linking specific movements (like tail oscillation) to changes in CSF pressure.

My second biggest complaint (which is somewhat intertwined with the first) relates to the fact that ICP studies in ambulatory humans are not cited and the authors, particularly in paragraph #3 of page 2 (Intro) and in the Discussion section, suggest to the readers that such data does not exist.  As a practicing pediatric neurosurgeon I not uncommonly do ambulatory ICP monitoring with patients, particularly those patients suspected of having idiopathic intracranial hypertension that do not have papilledema on fundoscopic exam.  Based on this personal experience I very much believe all the data presented as it agrees with clinical experience in humans. So, I think the authors need to include some more references about the human data ICP data that has been published.  There are many examples but here are a couple that come to mind:

When I first drafted the manuscript, I agonized over how to treat the telemetric studies I was familiar with. On the one hand, many/most of them show patterns of change in CSF pressure that have similar temporal and amplitude patterns to what we found in our study. On the other hand, the studies that I am familiar with provide very little detail about exactly what the patient is doing during these episodes of CSF pressure elevation. The animal studies were easier to treat, since they all published ICP (or CSF pressure) graphs with clearly marked "movement" periods. I did not want to appear to criticize the telemetric literature for failing to identify the exact movements of the patient, when, after all, the entire purpose of the telemetry is to provide some "freedom" to the patient.

That being said, the Reviewer is correct, in pondering this I failed to do justice to a very pertinent body of literature. I have corrected this omission, and noted the relevance of telemetric studies in both the introduction and the discussion.

1) Telemetric ICP monitoring systems exist, see:

Tschan CA, Velazquez Sanchez VF, Heckelmann M, Antes S. Home telemonitoring of intracranial pressure. Acta Neurochir (Wien). 2019 Aug;161(8):1605-1617. doi: 10.1007/s00701-019-03959-5. Epub 2019 Jun 5. PMID: 31168730

2) There are numerous studies in humans, including those without CSF disorders, looking at ICP recordings in mobile individuals.  Here is one that involved ICP monitoring during a mountain climbing expedition as an extreme example:

Wilson MH, and Milledge J. (2008). Direct measurement of intracranial pressure at high altitude and correlation of ventricular size with acute mountain sickness: Brian Cummins’ results from the 1985 Kishtwar expedition. Neurosurgery 63:970–974; discussion 974–975.

My third thought when reading this as a non-alligator expert was that it would have been nice to have a little primer on unique anatomic features of the alligator relevant to this study in the introduction.  For example the lack of a caudal cistern and presence of a prominent lumbar cistern in the alligator (mentioned lines 477-479 in Discussion) would have been helpful to know about prior to interpreting the results.

I agree completely; however, the journal has a limit for self-citations. Not surprisingly, my lab is the only lab publishing studies of CSF pressure in Alligator, and one of the most active labs publishing on related issues such as heart rate, anesthesia, etc. To keep under the self-citation limit I had to avoid citing some of my own previous studies of CSF dynamics in Alligator. I could not summarize the interesting aspects of the CNS and meninges of Alligator, without multiples references to my own work, and I simply do not have the citation "flexibility" to do that.

Highly specific notes:

- Line 40: It pains me that hydrocephalus, the canonical abnormal CSF disorder is not mentioned.  I would personally recommend adding it to the list of syringomyelia and Alzheimer's

Thank you for catching that omission, it was corrected.

- Line 97: "fit" should be "fitted"

Indeed it should, the suggested change was made to the manuscript.

- Figure 10: The figure legend is inadequate.  The body movements that produced the A and B recordings should be described.

I revised all of the figures for clarity and information content. Along with them I modified almost all of the figure legends in an attempt to do a better job of guiding the reader to the key points of each figure.

- The discussion section gets repetitive.  In particular please note the similarity in content between lines 412-413 & lines 425-427

I trimmed down the discussion, and, in particular, tried to eliminate any repetition (including the passages you correctly flagged).

Reviewer 2 Report

See attached file.

Author Response

This is an interesting paper that provides new data describing how body movements can have a large effect on CSF pressures, using alligators as a model. It requires some further refinement, mostly in the reporting of the methods and results as follows.

Major Comments:

  1. 46 In humans at least, cardiac pulsations are important in the spinal canal, not just respiratory-linked CSF movements. Please correct. There are several recent human PC-MRI studies of CSF flows in the spinal canal that are more up-to-date and accurate references than ref [8].

I appreciate the Reviewer’s comment. Somewhat as an aside, I am amazed at how “contentious” the relative importance of cardiac vs. respiratory / flow vs. pressure / skull vs. spinal cord remains. In the review process of my last manuscript I had one reviewer arguing that the experimental evidence clearly demonstrated the importance of cardiac-related pulsation to CSF dynamics in the skull, while the other reviewer insisted that this be removed from the manuscript since all the evidence indicated that the cardiac-pulsation had no importance to CSF dynamics. In evaluating how best to respond to Reviewer 2, I decided that the sentence in question (and reference [8]) was not really an integral part of the introduction, so I simply deleted the entire sentence.

  1. 54 This comment about validity of CSF flow studied by MRI being questionable is a gross over-simplification. PCMRI studies that measure CSF velocities are well established and validated. There remains some debate about whether calculating net flows from these large pulsatile motions is accurate. Please correct.

The Reviewer is absolutely correct. The phrase in question was poorly worded and cast too broad of a net. I have reworked this portion of the Introduction.

  1. 57 The authors omit any discussion here (or in the discussion section) of how implanted pressure transducers can disturb CSF dynamics, both by their presence in a very narrow CSF space, but also by altering fluid pressures as a result of CSF leaks during implantation. Please adjust both introduction and discussion, and also mention how this was managed in the methods (~line 130).

Respectfully, I must disagree. I use a variation of the classic EVD approach in which my pressure catheter touches the (deep) surface of the dura without extending into the CSF. This is done specifically to minimize any impact on the CSF dynamics. This is all done under a surgical microscope, and can be done with absolutely no loss of CSF. The dura is then double-sealed around the catheter (once with surgical adhesive, then again with epoxy). The experiments presented in this study were performed with no leakage of CSF.

At the same time, I do recognize that this is an important point, so I have modified the discussion to include this information.

  1. 60-74 The authors neglect to mention the importance of venous blood shifts between the thorax and abdominal cavities and the spinal canal as a result of pressures in those cavities which have been shown to play a significant role in CSF movements in humans.

Absolutely. When I originally drafted those sentences I was (simplisitically) pooling the pressures in the body cavities and venous blood shifts together. The Reviewer is absolutely right that the latter deserve to be mentioned separately. This passage was modified in the manuscript.

  1. 166 Please use SI units (i.e. degrees Celsius not F)

Thank you for catching this...I am at a completely loss how it passed through rounds of revision! It was changed to C in the manuscript.

  1. 171-2 It is not clear why the majority were no used for data analysis. Please clarify if this is because the majority of the data were excluded for the reasons outlined in lines 173- 176

I apologize for any confusion. I reworded this section for clarity so the reader could understand exactly why any sequence was excluded from analysis.

  1. 177 Were 11 sequences analysed per animal, or 11 across all animals? If the latter, how many per animal? Later, the text implies data from only 3 animals is presented. Why, when seven animals were studied? This needs clarification.

I am not sure what led the Reviewer to think that data from only three animals is presented. In the figures presented where overlapping data lines were used we found that presenting more than three data traces cluttered up the figure too much and detracted from the point we were trying to make, so we generally placed three data traces onto each figure. But that does not mean that we only have three data sets.

That being said, I have added an explicit statement to the end of the materials and methods regarding the distribution of data from the different alligators.

  1. 187 Please explain the causes of these apneas – were they ‘natural’ or induced experimentally?

The apneas presented are all induced...we simply turn off the ventilator...since we are monitoring the exhalatory gases we know that this simple approach produces a true apnic condition. The text has been modified to clarified that these are induced apneas.

  1. All figs. For all graphs, please include legends explaining the various colors on the graphs, and put the variable(s) being measured on the y axis titles. In some figures, the lines should be thicker for legibility.

The figures were modified for clarity and increased information content.

  1. All figs. For all figures where specific movements/events occur, these should be more clearly indicated on the plots, perhaps by superimposing a bar for the duration of the event, labelled with the event type on the plot.

Upon reflection, I agree and the suggested change was made in each figure.

  1. 192 The average heart rates reported in the caption of Fig 1 are inconsistent with those reported in the main text.

I am not sure what inconsistency the Review is referencing. The Figure 1 caption notes that cardiac dominant frequencies from the three traces shown are "near 0.4 Hz" which corresponds to heart rate of 24 beats per minute. In the figure two of the traces have a dominant peak slightly below the vertical arrow (which is at 0.4 Hz), one trace is slightly higher. The text indicates that the mean from all seven alligators was 0.37 HZ which corresponds to 22 beats per minute. Alligators, like people have variation in their heart rates. I apologize, as I am truly not trying to flippant, but I am not sure where the "inconsistency" is?

  1. 220 It is unclear why a few of the heart rate measures are reported with standard error, when other variables are reported with standard deviation, even though they are all averages of averages as far as I can tell.

Thank you for catching that, the s.e. was purely a typo, they were all s.d. This error has been corrected in the manuscript.

  1. Fig 3 It is potentially misleading to plot these pressures offset by 5mmHg. Please use different line styles with a consistent pressure scale, or plot on four separate panels if that doesn’t allow different traces to be visualised

The traces were plotted with an offset in an attempt to make it easier for the reader to follow each line. I take the Reviewer's point, and certainly don't want to mislead the reader. The figure was reworked with the four traces plotted together.

  1. 267-268 This observation is not clear from Figure 6

The figure, and figure legend, were modified to improve clarity and increase information content.

  1. 294-297 It is not clear how these statistics were done. What test was used? How was the repeated measures within an animal accounted for?

These are all simple paired t-test. That information was added to the manuscript.

  1. Fig 8 Needs legend explaining the colours (different animals perhaps?)

The legends for all of the figures with multiple traces were modified to indicate that the different colors do, in fact, represent traces from different alligators.

  1. 323-324 The concept of ‘body tone’ is introduced here with no definition and no methods describing how it was measured. It’s referred to again lines 380-382.

The text has been modified to try to be more explicit about what is meant by body tone, and how we could identify changes in body tone.

  1. 361 It is unclear what this ‘percent scale’ is – percent of what?

The figure legend was modified to provide this information...the figure is a standard "box plot".

  1. Table 1 This table should include both baseline (mean) CSF pressures and the amplitude of the pulses/variation.

Respectfully, we disagree. Most of the movements we are describing only influenced the amplitude of the CSF pulses, not the baseline. As such the new column in the table would be the same for most variables. Two of the movements where the baseline shifted (active ventilation and body tone) both positive and negative shifts were observed (in nearly equal number and magnitude, as show in Figure 10)...so a simple summary  number may actually mislead the reader. The locomotion is the only movement in which the baseline had a consistent pattern of shift (upward). We feel that is presented clearly enough in the text and Figure 12C that it does not warrant a separate column in Table 1.

  1. 466-473 The explanation/discussion of the CSF pressures and their relationship to movements Is vague and unclear. The authors should discuss fluid velocity/pressure relationship here more clearly, as the sensor is measuring pressure, not momentum. Ideally, bringing in the relevant fluid mechanics laws rather than making vague statements about fluid momentum without linking to either velocity or pressure.

I appreciate the Reviewer’s point, but I respectfully disagree. These experiments were undertaken will the purpose of documenting the influence of movement on CSF pressure. We are explicitly not trying to provide a complete fluid mechanics explanation of what we observed. Frankly, I don’t feel we could; in part this stems from the fact that we only have pressure (not velocity) data, but also because for some movements (like displacement of the limb) we can’t be positive about the functional/mechanical link between the movement and CSF pressure. Trying to interpret the different movements in terms of formal fluid mechanics would bog down the discussion without, I believe, adding much clarity. Our purpose for this paper was to demonstrate to the reader that: 1) movement altered the CSF pressure; and 2) there was considerable variation in terms of the relative impact on CSF pressure of the different movements. I think we can more cleanly accomplish those two goals without getting into formal fluid mechanics.  Mind you, I agree with the Reviewer on the value/clarity of the fluid mechanics perspective; we just completed another round of experiments designed specifically to enable us to treat the alligator’s CSF in a more formal/rigorous fashion.

  1. Discussion A comprehensive discussion of the limitations in this study is lacking in the discussion.

As part of the revision I incorporated a new paragraph in the discussion to address some of these issues.

Minor Comments:

  1. all CSF should be capitalised not lower case (i.e. CSF not csf) in the text.

I agree, I am not sure how/why it was in lower case in the introduction then capitalized elsewhere; this has been fixed.

  1. all Some of the language could be further refined. E.g. arterial pressure rather than arteriole pressure, use of respiratory cycle rather than ‘ventilator cycle’ and similar places except when the animal is being mechanically ventilated. ‘Alzheimer’s Disease’ not “Alzheimer’s”

I have no problem with changing arteriole to arterial. The use of “ventilation” is different…ventilation is the correct term for what we are looking at, and in the alligator ventilation (the mechanical movement of air) can be/is decoupled from respiration (gas exchange with the blood) so that the distinction is more important than it is in humans. That being said, I understand that “respiration” is used in a broader sense in most human studies. Since ventilation is the correct term for what I am describing, and since I think it would create potential confusion to switch back and forth, I think it best to use ventilation.

  1. Reference formatting has gone awry – most references have the number twice.

I numbered the references as required by the journal, which then added numbers when they made the review proof. There will only be one set of numbers in the published paper.

  1. 76 ‘Fictive’ is not the right word here, perhaps ‘externally imposed’ or ‘passive’

With all due respect to the Reviewer, I must disagree. There is a large body of literature on “Fictive” locomotion and body movement. It is the accepted term for artificially-induced body movement.

  1. 107 Fix grammar

This error was corrected; thank you for bringing it to our attention.

  1. 216- 218 Some units are missing from here. E.g. should be ‘30 s’ in a couple of places

I apologize for any confusion this created. For some reason when the Review pdf was created the degree symbols (°) were dropped. They were present in the manuscript and will be there in the published paper.

  1. 244 Suggest ‘Expanded time scale’ rather than ‘finer resolution’. It would also be good to indicate which region of A has been expanded.

I agree, “Expanded time scale” is clearer; that change was made. (B) is on an expanded time scale, but it is from a different data trace than (A)…the figure never never stated that it was the same trace. Clearly the same high-frequency pulsations are evident in (A).

  1. 251 Perhaps ‘regular distance’ could be changed to ‘constant distance’ for clarity

I agree and thank the Review for this helpful suggestion.

  1. 355 Remove period in the middle of the first sentence

Thank you for catching this error, it was corrected.

  1. 392 Fix grammar

Thank you for catching this error, it was corrected.
